# Personalized Federated Adaptation via Prototype–Text Contrastive Alignment

## Abstract

Personalized federated learning (PFL) aims to share global knowledge while tailoring models to heterogeneous clients. However, traditional PFL methods face two key challenges: (i) reliance on aggregation-based model updates to form shared information makes training sensitive to data heterogeneity; and (ii) repeated on-client training and transmission of full model parameters or gradients yield substantial computational and communication overhead. Inspired by vision–language models (VLMs) such as CLIP, we pursue an alternative paradigm that adapts a frozen backbone through lightweight modules that leverage language-anchored priors. Specifically, we propose Personalized Federated Adaptation via Prototype–Text Contrastive Alignment (FedPACT), which treats a client-specific personalized prototype cache and a shared text head as the only trainable and communicated components. Clients update only their prototypes to fit local distributions, while the server refines the shared text head by contrastively aligning the text embeddings with personalized prototypes. Our theoretical analysis shows that the shared text head improves convergence of the personalized prototype cache by enlarging the prototype–text margin. Experiments demonstrate that FedPACT achieves superior personalized and global performance over state-of-the-art methods.

## 1 Introduction

Federated learning (FL) (McMahan et al., 2017; Singh et al., 2022) enables collaborative model development across distributed data silos while keeping data local. In many practical deployments, however, client data exhibit pronounced non-independent and identically distributed (non-IID) characteristics arising from domain shift, feature distribution mismatch, label imbalance, and even missing classes on subsets of clients (Zhang et al., 2022a; Wang et al., 2024). These forms of data heterogeneity degrade the utility of a single global model and can yield per-client performance drops, i.e., negative transfer, when the global distribution is misaligned with local distributions (Wang et al., 2020; Acar et al., 2021; Wang et al., 2024). Personalized federated learning (PFL) formulates collaboration across heterogeneous clients as a multi-task learning problem, jointly optimizing related yet distinct client models via (i) regularization based coupling (Smith et al., 2017; Deng et al., 2020; T Dinh et al., 2020; Huang et al., 2021), (ii) meta learning (Jiang et al., 2019; Fallah et al., 2020), (iii) model decomposition (Collins et al., 2021; Liang et al., 2020; Oh et al., 2021), and (iv) group-aware collaboration (Ghosh et al., 2020; Li et al., 2023; Lu et al., 2023), thereby mitigating data heterogeneity. Despite their effectiveness, most PFL methods still face two key challenges. First, data heterogeneity can cause shifts in local models, with the extent of these shifts increasing with the depth of the model (Luo et al., 2021). As a result, some PFL methods that aggregate client gradients or model parameters may produce suboptimal shared information, making the model performance highly sensitive to data heterogeneity (Wang et al., 2020; Acar et al., 2021; Wang et al., 2024). Second, most PFL methods still involve iterative on-client training of sizable local/shared components and repeated exchange of parameter updates for the shared part across rounds, leading to nontrivial computation and communication overhead (Lee & Lee, 2021; Yang et al., 2023).

Recent progress in Vision–Language Models (VLMs) (Jia et al., 2021; Kim et al., 2021) offers an opportunity to rethink traditional PFL frameworks. Models such as CLIP (Radford et al., 2021) are trained at web scale and provide robust and transferable visual representations paired with language-grounded text embeddings. Notably, VLMs expose a semantic interface through prompts and class

names that can steer prediction without modifying the visual backbone. This property suggests a paradigm shift for PFL: rather than repeatedly retraining and transmitting large models, it is possible to adapt a frozen VLM using small modules that exploit pretrained text features as stable, cross-client semantic anchors (Zhang et al., 2022b). Under non-IID data, such anchors help counter client drift by aligning local decision boundaries to a common language-informed space, and they mitigate missing-class scenarios by providing meaningful targets even where local examples are scarce.

Guided by these insights, we introduce Personalized Federated Adaptation via Prototype–Text Contrastive Alignment (**FedPACT**), which freezes the pretrained image encoder and divides the learnable components into two lightweight modules to address the challenges of high communication and computation costs. Specifically, visual features are generated by the frozen image encoder and processed by two trainable components: the shared text head, which provides language-grounded class directions and outputs scores aligned with global semantics, and the personalized prototype cache, which computes affinities between the input and stored prototypes to generate client-specific scores. These scores are then combined via a residual connection to produce the final prediction. To mitigate the sensitivity of parameter aggregation to data heterogeneity, we propose prototype–text contrastive alignment to update the shared text head. The server uses contrastive learning to update the shared text head, treating each personalized prototype with its class text embedding as a positive example and pairing it with other class texts as negatives. On the client side, the shared text head remains fixed while each client optimizes its personalized prototype cache using local data, adapting decision boundaries to the local distribution while maintaining alignment with the shared text head. Repeating these steps enables the shared text head to capture global information and remain robust to non-IID data, while the personalized prototypes effectively adapt to local data, ensuring personalized performance.

**Contributions.** We summarize our contributions as follows:

- We propose a novel VLM-based PFL framework that freezes the pretrained image encoder and personalizes a lightweight client-side prototype cache, with a server-side shared text head facilitating cross-client knowledge transfer and aggregation.

- We design an alternating prototype–text contrastive alignment scheme. On clients, the shared text head remains fixed while each client optimizes its personalized prototype cache using local data, adapting decision boundaries to its distribution. On the server, contrastive learning updates the shared text head using the personalized prototypes, steering it toward directions supported across clients.

- We establish a margin-controlled local convergence bound, demonstrating that enlarging the feature–text margin accelerates convergence. We further prove that contrastive updates to the shared text head enlarge the global prototype–text margin, and this gain transfers to the local prototype–text margin when the personalized prototypes do not significantly deviate from the global prototypes, thereby improving local convergence.

## 2 RELATED WORK

### 2.1 PERSONALIZED FEDERATED LEARNING

FL enables collaborative model training across a central server and multiple clients without sharing raw data. However, pronounced data and system heterogeneity often render a single global model inadequate. PFL reframes FL as multi-task optimization that jointly learns related yet distinct client models while exploiting cross-client commonality. Representative directions are as follows. First, regularization based methods couple client objectives through shared constraints or priors so that models share a common structure while allowing client-specific deviations, balancing transfer and personalization with convergence guarantees (Smith et al., 2017; Deng et al., 2020; T Dinh et al., 2020; Huang et al., 2021). Second, meta learning based methods learn an initialization or proximal prior, e.g., MAML-style meta-initialization, that enables rapid on-client adaptation (Jiang et al., 2019; Fallah et al., 2020). Third, model structure based methods either share a global backbone while personalizing lightweight local layers (Collins et al., 2021; Liang et al., 2020; Oh et al., 2021) or learn layer-wise aggregation weights that leverage inter-user similarity for personalized layer-wise aggregation (Ma et al., 2022). Fourth, group-aware collaboration discovers latent client groups and trains per-group models via data-distribution-based or game-theoretic clustering. (Ghosh et al.,

2020; Li et al., 2023; Lu et al., 2023). Despite their effectiveness, most PFL methods still require iterative training of sizable local or shared components and repeated transmission of shared parameters across rounds, incurring substantial computation and communication costs. Recent studies therefore integrate VLMs to leverage frozen pretrained backbones and confine personalization to lightweight modules (Lu et al.; Ghiasvand et al., 2025), thereby alleviating on-client computation and communication costs.

## 2.2 Federated Learning with VLMs

VLMs such as CLIP are trained at web scale on paired image–text data and learn aligned, language-grounded embeddings that transfer robustly across tasks via promptable interfaces, which makes them attractive foundations for FL where data heterogeneity, and resource constraints dominate. Freezing VLM backbones while adapting only small, semantics-aware modules enables FL systems to reduce computation and communication cost and to stabilize collaboration via a shared language-grounded prior. Recent work on FL with VLMs falls into three families. First, prompt-tuning methods learn textual prompts or prompt generators to steer a frozen VLM, where FedTPG employs text-conditioned prompt generation to improve generalization to unseen classes and datasets (Ghiasvand et al., 2025; Guo et al., 2023; Qiu et al., 2024). Second, LoRA/adapter-based PEFT replaces full-model updates with low-rank or adapter inserts that are the only communicated/optimized parameters (Sundaram et al., 2019; Zhang et al., 2022b; Han et al., 2024), improving efficiency under data heterogeneity. For instance, TriplePlay (Imteaj et al., 2024) combines LoRA with quantization and long-tail handling to accelerate convergence and reduce bandwidth; FedCLIP (Lu et al.) attaches attention-based adapters to retain model generality while support local adaptation; pFed-MMA (Ghiasvand et al., 2025) introduces multi-modal adapters with a globally shared projection to balance personalization and global generalization; FAA-CLIP (Wu et al., 2025) trains a lightweight feature adaptation module with domain-adversarial alignment for cross-domain robustness. Third, cache-based methods construct a key–value cache from the few-shot training set and adapt CLIP via feature retrieval. For instance, CacheFL (Yi et al., 2025) initializes a class-balanced cache via generative synthesis and aggregates only cache parameters, while FedPGA (Liu & Huang, 2025) employs prototype-guided cache to mitigate inter-client shifts.

Note that prompt-tuning–based methods still require backpropagation through the entire model during local training, imposing nontrivial client-side compute and memory cost. LoRA/adapter-based PEFT and cache-based methods, especially VLM-based PFL approaches such as FedCLIP and pFedMMA, still rely on gradient aggregation of local modules and can inherit sensitivity to non-IID data. In contrast, our proposed FedPACT replaces weight averaging with contrastive alignment between client-side personalized prototype caches and the server-side shared text head, improving robustness to data heterogeneity while maintaining low computation and communication overhead.

## 3 Methodology

### 3.1 Preliminary

Consider a PFL scenario with $K$ clients indexed by $\mathcal{K} = \{1, 2, \ldots, K\}$. Each client $k$ holds a local dataset $\mathcal{D}_k = \{(\mathbf{x}_i, y_i)\}_{i=1}^{m_k}$ with class set $\mathcal{Y} = \{1, \ldots, N\}$. A personalized federated objective can be written as

$$\min_{\{\mathbf{w}_k\}, \phi} \sum_{k \in \mathcal{K}} \pi_k \left[ \mathbb{E}_{(\mathbf{x}, y) \sim \mathcal{D}_k} \left[ \ell\big(S(\mathbf{x}; \mathbf{w}_k, \phi), y\big) \right] + \gamma \Omega\big(\mathbf{w}_k, \phi\big) \right], \tag{1}$$

where $S(\cdot)$ is the scoring function, $\ell(\cdot)$ is the classification loss, $\mathbf{w}_k$ are client-specific parameters, $\phi$ are shared parameters that couple clients, $\Omega$ penalizes deviation from the shared representation, and $\pi_k = \frac{m_k}{\sum_{k \in \mathcal{K}} m_k}$. In this work, we consider PFL for VLMs with a pretrained image encoder, where each input $x$ is mapped to a unit-norm feature $\mathbf{f} = f(\mathbf{x}) \in \mathbb{R}^C$ with embedding dimension $C$. Inspired by (Zhang et al., 2022b), personalization is realized by a client-side personalized prototype cache $\mathbf{V}_k \in \mathbb{R}^{N \times C}$, while language-grounded semantics are encoded by a server-side shared text head $\mathbf{H} \in \mathbb{R}^{N \times C}$. The class logits are decomposed into a cache term and a text term:

$$S(\mathbf{f}; \mathbf{V}_k, \mathbf{H}) = \alpha \varphi\big(\mathbf{f}\mathbf{V}_k^\top\big) + \mathbf{f}\mathbf{H}^\top, \tag{2}$$

where $\varphi(z) = \exp[-\beta(1 - z)]$ with $\alpha, \beta > 0$.

Figure 1: Overview of the FedPACT framework. Different shapes denote the personalized prototype cache and the shared text head, while colors indicate different classes. Initialization: Each client constructs local prototypes and uploads them to the server; the server aggregates these local prototypes to initialize the personalized prototype cache and, in parallel, initializes the shared text head from class names using a text encoder. Training: The server broadcasts the shared text head, which is kept frozen on each client. Clients then update their personalized prototype cache using local features (see equation 5) and upload the updated cache to the server. The server refines the shared text head by contrastively aligning it with the received prototype caches (see equation 8), thereby completing one communication round.

## 3.2 ALTERNATING PROTOTYPE–TEXT CONTRASTIVE ALIGNMENT

The alternating optimization over $\mathbf{V}_k$ and $\mathbf{H}$ implements a bi-level scheme: clients minimize empirical risk with respect to $\mathbf{V}_k$, and the server updates $\mathbf{H}$ by a contrastive objective constructed from personalized prototypes. We detail the three building blocks below.

**Initialization.** At round $t = 0$, each client computes local prototypes by averaging normalized features:

$$\widehat{\mathbf{v}}_{k,n} = \frac{1}{|\mathcal{D}_{k,n}|} \sum_{(\mathbf{x},y) \in \mathcal{D}_{k,n}} f(\mathbf{x}), \tag{3}$$

where $n \in \mathcal{Y}$, $\mathcal{D}_{k,n} = \{(\mathbf{x}, y) \in \mathcal{D}_k : y = n\}$, and uploads the local prototypes to the server. The server aggregates the local prototypes to form global prototypes

$$\overline{\mathbf{v}}_n = \frac{\sum_{k \in \mathcal{K}} |\mathcal{D}_{k,n}| \widehat{\mathbf{v}}_{k,n}}{\sum_{k \in \mathcal{K}} |\mathcal{D}_{k,n}|}, \tag{4}$$

and broadcasts $\overline{\mathbf{V}} = [\overline{\mathbf{v}}_1; \ldots; \overline{\mathbf{v}}_N]$ to all clients. Then each client initializes its cache as $\overline{\mathbf{V}}$ and the server initializes the shared text head by encoding the class names $c_n$ with a text encoder $\mathbf{h}_n^{(0)} = h(c_n)$, and $\mathbf{H}^{(0)} = \mathbf{H}_{\text{CLIP}} = [\mathbf{h}_1^{(0)}; \ldots; \mathbf{h}_N^{(0)}]$.

**Client-Side Prototype Personalization.** At communication round $i$, each client $k$ receives the current shared text head $\mathbf{H}^{(i)}$ and updates only its personalized prototype cache $\mathbf{V}_k$ by minimizing the local objective

$$\min_{\mathbf{V}_k} \mathcal{L}_k \left( \mathbf{V}_k; \mathbf{H}^{(i)}, \mathcal{D}_k \right) = \mathbb{E}_{(\mathbf{x},y) \sim \mathcal{D}_k} \left[ \ell_{\text{CE}} \left( S \left( f(\mathbf{x}); \mathbf{V}_k, \mathbf{H}^{(i)} \right), y \right) \right], \tag{5}$$

where $\ell_{\text{CE}}$ denotes the standard cross-entropy loss. During local training, the shared text head remains frozen and serves as a fixed semantic reference, keeping local updates aligned with the common language-grounded space. After completing local training, each client uploads the updated $\mathbf{V}_k$ to the server.

---

**Algorithm 1** Alternating Prototype–Text Contrastive Alignment

---

**Require:** Client set $\mathcal{K}$, client datasets $\{\mathcal{D}_k\}$, rounds $I$, local epochs $T$, frozen encoder $f(\cdot)$, initial text head $\mathbf{H}^{(0)} = \mathbf{H}_{\text{CLIP}}$.

1: # Initialization ($i = 0$)
2: **for** each client $k$ **in parallel do**
3:     compute local prototypes $\widehat{\mathbf{v}}_{k,n}$ and upload the local prototypes to the server
4: **end for**
5: **Server:** compute global prototypes $\overline{\mathbf{v}}_n$ and set $\overline{\mathbf{V}} = [\overline{\mathbf{v}}_1; \dots; \overline{\mathbf{v}}_N]$; broadcast $\overline{\mathbf{V}}$ to all clients

6: **for** $i = 1 \dots I - 1$ **do**
7:     # Client step
8:     **for** each client $k$ **in parallel do**
9:         update $\mathbf{V}_k$ by minimizing $\mathcal{L}_k\left(\mathbf{V}_k; \mathbf{H}^{(i)}, \mathcal{D}_k\right)$         ▷ see equation 5
10:       upload updated $\mathbf{V}_k^{(i)}$ to the server
11:     **end for**
12:     # Server step
13:     construct $\mathcal{S}^{(i)} = \left\{\left(\mathbf{v}_{k,n}^{(i)}, n\right)\right\}$ from received $\mathcal{V}^{(i)} = \left\{\mathbf{V}_k^{(i)}\right\}$
14:     update $\mathbf{H}$ by minimizing $\mathcal{L}_s(\mathbf{H}; \mathcal{S}^{(i)})$         ▷ see equation 8
15:     broadcast updated $\mathbf{H}^{(i+1)}$ to all clients
16: **end for**

---

**Server-Side Contrastive Text Adaptation.** At communication round $i$, the server receives personalized prototype caches from clients $\mathcal{V}^{(i)} = \{\mathbf{V}_k^{(i)}\}_{k \in \mathcal{K}}$ and construct a labeled prototype set $\mathcal{S}^{(i)} = \{(\mathbf{v}_{k,n}^{(i)}, n) \mid k \in \mathcal{K}, \ n \in \mathcal{Y}\}$, where $\mathbf{v}_{k,n}^{(i)}$ is the class-$n$ prototype embedding given by the $n$-th row of $\mathbf{V}_k^{(i)}$. The shared text head $\mathbf{H}$ with rows $\{\mathbf{h}_n\}_{n \in \mathcal{Y}}$ is then updated by minimizing a contrastive InfoNCE loss (Oord et al., 2018) that pulls matched text–prototype pairs $(\mathbf{h}_n, \mathbf{v}_{k,n}^{(i)})$ together while pushing the same prototype away from all non-matching class texts $\{(\mathbf{h}_j, \mathbf{v}_{k,n}^{(i)})\}_{j \neq n}$:

$$\mathcal{L}_{\text{align}}(\mathbf{H}; \mathcal{S}^{(i)}) = \sum_{k \in \mathcal{K}} \frac{\pi_k}{N} \sum_{n \in \mathcal{Y}} \left[ -\log \frac{\exp\left(\mathbf{h}_n^\top \mathbf{v}_{k,n}^{(i)}/\tau\right)}{\sum_{j=1}^N \exp\left(\mathbf{h}_j^\top \mathbf{v}_{k,n}^{(i)}/\tau\right)} \right], \tag{6}$$

with temperature $\tau > 0$. To limit semantic drift from the original VLM space, we add a stability regularizer:

$$\mathcal{L}_{\text{reg}}(\mathbf{H}) = -\frac{1}{N} \sum_{n=1}^N \left\langle \mathbf{h}_n, \mathbf{h}_n^{(0)} \right\rangle. \tag{7}$$

In summary, the server solves

$$\min_{\mathbf{H}} \mathcal{L}_s(\mathbf{H}; \mathcal{S}^{(i)}) = \mathcal{L}_{\text{align}}(\mathbf{H}; \mathcal{S}^{(i)}) + \lambda_{\mathbf{H}} \mathcal{L}_{\text{reg}}(\mathbf{H}), \tag{8}$$

with $\lambda_{\mathbf{H}} \geq 0$ and broadcasts the updated $\mathbf{H}$ to all clients to complete the round one communication round.

**Summary.** The overall procedure is summarized in Algorithm 1. Repeating equation 5 and equation 8 yields progressive prototype–text contrastive alignment: the shared text head evolves into language-anchored vectors that capture global information and remain robust under non-IID variability, while personalized prototype caches adapt decision boundaries to local distributions. The scheme requires no backbone retraining and communicates only prototypes, offering both computation and communication efficiency.

# 4 THEORETICAL ANALYSIS

In this section, we provide a theoretical analysis to better understand how the shared text head $\mathbf{H}$ controls the convergence of personalized prototype cache updates. Our analysis is based on the following assumptions:

**Assumption 1.** *(Smoothness) For each client $k$ and round $t$, the local loss $\mathcal{L}_k(\cdot)$ is $L_{\mathbf{V}}$-smooth in $\mathbf{V}$, i.e., for all $\mathbf{V}, \mathbf{V}'$, $\left\| \nabla_{\mathbf{V}} \mathcal{L}_k(\mathbf{V}) - \nabla_{\mathbf{V}} \mathcal{L}_k(\mathbf{V}') \right\| \leq L_{\mathbf{V}} \|\mathbf{V} - \mathbf{V}'\|$.*

**Assumption 2.** *(Bounded Gradient Variance) For each client $k$, the mini-batch stochastic gradient $g_k$ computed with batch size $b$ has bounded variance: $\mathbb{E}\left[ \left\| g_k - \nabla_{\mathbf{V}} \mathcal{L}_k(\mathbf{V}_k) \right\|^2 \right] \leq \frac{\sigma_c^2}{b}$.*

Given a shared text head $\mathbf{H} = [\mathbf{h}_1; \ldots; \mathbf{h}_N]$, define the margin induced by $\mathbf{H}$ for a feature $\mathbf{f}$ with label $y$ as

$$m_{\mathbf{H}}(\mathbf{f}, y) := \mathbf{f}^\top \mathbf{h}_y - \max_{j \neq y} \mathbf{f}^\top \mathbf{h}_j. \tag{9}$$

Then, we obtain the margin-controlled local convergence, as stated in the following theorem:

**Theorem 1.** *When Assumptions 1–2 hold, given the shared text head $\mathbf{H}$, number of SGD steps $T$ and the learning rate $\eta \leq \frac{1}{L_{\mathbf{V}}}$, the convergence rate of personalized prototype cache $\mathbf{V}_k$ is given by*

$$\min_{0 \leq t < T} \mathbb{E}\left[ \|\nabla_{\mathbf{V}} \mathcal{L}_k(\mathbf{V}_k^{(t)})\|^2 \right] \leq \frac{2\left(\mathcal{L}_k(\mathbf{V}_k^{(0)}) - \mathcal{L}_k^\star\right)}{\eta T} + \eta L_{\mathbf{V}} (\alpha\beta)^2 \frac{(N-1)}{b} \mathbb{E}_{(\mathbf{f}, y) \sim \mathcal{D}_k}\left[ e^{-m_{\mathbf{H}}(\mathbf{f}, y)} \right], \tag{10}$$

*where $\mathcal{L}_k^\star := \inf_{\mathbf{V}} \mathcal{L}_k(\mathbf{V})$ is the optimal objective value for client $k$.*

Theorem 1 highlights the role of the shared text head, which can tighten the local convergence bound by enlarging local margins $m_{\mathbf{H}}(\mathbf{f}, y)$, thereby improving the local convergence performance. Therefore, we analyze the effect of the server-side contrastive text adaptation, i.e., the update of the shared text head $\mathbf{H}$, specified in equation 6. Define the global margin as the sum of margins over all personalized prototypes as

$$M(\mathbf{H}) = \sum_{k \in \mathcal{K}} \pi_k \sum_{n \in \mathcal{Y}} m_{\mathbf{H}}(\mathbf{v}_{k,n}, n), \tag{11}$$

we obtain the following Lemma.

**Lemma 2.** *If the update in equation 6 strictly decreases the contrastive objective, i.e., $\mathcal{L}_{align}(\mathbf{H}^{(t+1)}) < \mathcal{L}_{align}(\mathbf{H}^{(t)})$, the global margin satisfies*

$$M\left(\mathbf{H}^{(t+1)}\right) > M\left(\mathbf{H}^{(t)}\right). \tag{12}$$

Lemma 2 establishes a strict increase in the global margin $M(\mathbf{H})$, indicating that the shared text head $\mathbf{H}$ is progressively aligned with directions supported by personalized prototypes. Building on Lemma 2, let $\bar{\mathbf{v}}_n$ denote the global prototype for class $n$ defined in equation 4, we have the following theorem.

**Theorem 3.** *If the update strictly decreases the contrastive objective, i.e., $\mathcal{L}_{align}(\mathbf{H}^{(t+1)}) < \mathcal{L}_{align}(\mathbf{H}^{(t)})$, then there exists $\varepsilon \in (0, 1)$ such that, for any local prototype $\mathbf{v}_{k,n}$ satisfying*

$$\frac{\mathbf{v}_{k,n}^\top \bar{\mathbf{v}}_n}{\|\mathbf{v}_{k,n}\| \|\bar{\mathbf{v}}_n\|} > \varepsilon, \tag{13}$$

*the local prototype–text margin strictly increases at the next iterate:*

$$m_{\mathbf{H}^{(t+1)}}\left(\mathbf{v}_{k,n}, n\right) > m_{\mathbf{H}^{(t)}}\left(\mathbf{v}_{k,n}, n\right). \tag{14}$$

Theorem 3 implies that the update direction steers each class-specific vector of the shared text head toward the subspace spanned by the federated centroids $\{\bar{\mathbf{v}}_n\}_n$. Consequently, if a client prototype $\mathbf{v}_{k,n}$ is sufficiently aligned with its centroid, i.e., $\frac{\mathbf{v}_{k,n}^\top \bar{\mathbf{v}}_n}{\|\mathbf{v}_{k,n}\| \|\bar{\mathbf{v}}_n\|} > \varepsilon$ for some $\varepsilon \in (0, 1)$, the global margin improvement established by Lemma 2 transfers to a strict increase in the local prototype–text margin, yielding $m_{\mathbf{H}^{(t+1)}}(\mathbf{v}_{k,n}, n) > m_{\mathbf{H}^{(t)}}(\mathbf{v}_{k,n}, n)$ and thereby enhancing local convergence. We will further investigate the effect of the alignment in our experiments.

Table 1: Test accuracy (%) for FedPACT and baselines with ResNet-50 backbone.

| Non-IID ($\rho$) | Method | Flower | DTD | Pets | Cars | UCF | Caltech | Food | SUN | Aircraft | EuroSAT | Mean |
|---|---|---|---|---|---|---|---|---|---|---|---|---|
| | CLIP-RN50 | 65.94 | 42.20 | 85.80 | 55.66 | 61.56 | 85.80 | 77.30 | 58.53 | 17.19 | 37.59 | 58.76 |
| $\rho = 0.1$ | FedPGA | 69.35 | 47.81 | **87.65** | 58.69 | 65.74 | 90.30 | **78.10** | 63.67 | 18.90 | 53.22 | 63.34 |
| | CacheFL | 67.48 | 45.69 | 87.49 | 57.49 | 64.34 | 89.90 | 77.84 | 61.85 | 18.15 | 47.19 | 61.74 |
| | **FedPACT-G** | **90.58** | **56.74** | 87.11 | **65.05** | **70.18** | **90.59** | 74.11 | **64.66** | **19.68** | **77.89** | **69.66** |
| | Tip-Adapter | 80.85 | 57.92 | **91.20** | 76.76 | 77.74 | 81.51 | 68.82 | 68.77 | 40.97 | 71.10 | 71.56 |
| | ECALP | 62.46 | 51.69 | 65.26 | 55.82 | 59.55 | 62.74 | 58.97 | 59.15 | 40.53 | 49.60 | 56.58 |
| | **FedPACT-P** | **94.57** | **72.01** | 90.08 | **79.11** | **81.73** | **95.15** | **79.30** | **77.93** | **42.29** | **83.74** | **79.59** |
| $\rho = 0.3$ | FedPGA | 72.59 | 51.06 | **87.74** | 61.46 | 67.17 | 90.39 | **78.32** | 65.44 | 19.74 | 56.73 | 65.06 |
| | CacheFL | 70.93 | 46.69 | 87.71 | 58.85 | 66.93 | **90.91** | 78.04 | 64.08 | 20.19 | 50.56 | 63.49 |
| | **FedPACT-G** | **90.34** | **57.80** | 87.38 | **65.30** | **71.27** | 90.43 | 75.52 | **65.66** | **21.75** | **78.16** | **70.36** |
| | Tip-Adapter | 71.63 | 47.72 | 88.32 | 67.90 | 70.83 | 79.17 | 63.10 | 59.53 | 30.11 | 63.32 | 64.16 |
| | ECALP | 39.74 | 36.40 | 44.52 | 40.57 | 39.43 | 50.37 | 45.65 | 42.06 | 27.51 | 44.28 | 41.05 |
| | **FedPACT-P** | **92.14** | **66.17** | **89.02** | **71.89** | **77.65** | **92.82** | **78.35** | **72.37** | **32.30** | **79.16** | **75.19** |

Table 2: Test accuracy (%) for FedPACT and baselines with ViT-B/16 backbone.

| Non-IID ($\rho$) | Method | Flower | DTD | Pets | Cars | UCF | Caltech | Food | SUN | Aircraft | EuroSAT | Mean |
|---|---|---|---|---|---|---|---|---|---|---|---|---|
| | CLIP-ViTB/16 | 71.38 | 44.39 | 89.07 | 65.27 | 66.75 | 92.94 | 86.11 | 62.55 | 24.84 | 47.77 | 65.11 |
| $\rho = 0.1$ | FedPGA | 74.18 | 48.94 | 48.94 | 68.03 | 72.80 | 94.69 | 86.21 | 68.73 | **29.55** | 60.02 | 69.36 |
| | CacheFL | 72.51 | 47.64 | 90.27 | 66.61 | 72.85 | **94.77** | **86.34** | 66.61 | 25.35 | 53.31 | 67.63 |
| | **FedPACT-G** | **94.23** | **61.23** | 91.88 | 73.67 | 78.19 | 94.60 | 85.56 | **70.17** | 27.51 | **80.78** | **75.78** |
| | Tip-Adapter | 84.82 | 56.80 | 93.48 | 83.43 | 83.61 | 81.31 | 74.14 | 70.53 | 50.13 | 77.61 | 75.59 |
| | ECALP | 62.96 | 51.93 | 65.02 | 62.84 | 63.48 | 63.43 | 64.03 | 62.68 | 47.10 | 53.30 | 59.68 |
| | **FedPACT-P** | **96.51** | **74.08** | **93.55** | **85.38** | **86.22** | **97.28** | **88.18** | **81.75** | **52.23** | **84.74** | **83.99** |
| $\rho = 0.3$ | FedPGA | 78.48 | 52.30 | 91.22 | 70.85 | 74.23 | 94.85 | 86.35 | 70.92 | 29.73 | 67.22 | 71.61 |
| | CacheFL | 75.72 | 48.05 | 91.20 | 67.68 | 75.18 | **95.25** | **86.50** | 68.99 | 28.05 | 63.69 | 70.03 |
| | **FedPACT-G** | **94.68** | **61.41** | **92.45** | **73.87** | **78.77** | 93.96 | 85.73 | **70.97** | **31.26** | **80.86** | **76.40** |
| | Tip-Adapter | 77.92 | 47.88 | 91.46 | 75.68 | 77.36 | 78.57 | 70.30 | 60.86 | 37.72 | 71.28 | 68.90 |
| | ECALP | 39.99 | 37.89 | 47.33 | 45.84 | 42.53 | 51.34 | 49.14 | 44.54 | 35.14 | 36.67 | 43.04 |
| | **FedPACT-P** | **95.17** | **68.11** | **93.65** | **80.07** | **82.18** | **95.84** | **87.40** | **76.74** | **42.91** | **82.39** | **80.45** |

## 5 EXPERIMENTS

### 5.1 EXPERIMENTAL SETTINGS

**Implementation Details:** Unless otherwise specified, we simulate a federated network with $K = 10$ clients. The client model adopts CLIP with ResNet-50 (He et al., 2016) and ViT-B/16 (Dosovitskiy et al., 2020) backbones, initialized from the official pretrained weights (Radford et al., 2021). We set the number of global rounds to $I = 10$. The personalized prototype cache is updated for $T = 20$ local epochs per round, while the shared text head is trained for $T = 100$ epochs at the server. We use the AdamW optimizer and set the mini-batch size to 16 for both the personalized prototype cache and the shared text head. The hyperparameters $\alpha$ and $\beta$, as well as the learning rates for both components, follow the settings in (Zhang et al., 2024). To induce label-distribution skew, we employ a Dirichlet allocator: for each dataset, per-client class proportions are sampled from $\mathrm{Dir}(\rho)$. Each client then draws examples up to its budget according to these proportions, yielding majority/minority class mixtures and potentially missing classes. Data heterogeneity is controlled by $\rho$, where a smaller $\rho$ indicates stronger non-IID.

**Baselines:** We evaluate accuracy under two testing protocols. (i) Global generalization: inference uses the server-trained shared text head (denoted **FedPACT-G**). As baselines, we include two cache- and VLM-based FL methods, FedPGA (Liu & Huang, 2025) and CacheFL (Yi et al., 2025). (ii) Personalization: inference uses each client's locally trained personalized prototype cache (denoted **FedPACT-P**). As baselines, we consider distributed variants of two few-shot vision–language adaptation methods, Tip-Adapter (Zhang et al., 2022b) and ECALP (Li et al.), both of which personalize on local data.

**Datasets:** We evaluate on ten widely used image classification datasets: Flowers102 (Nilsback & Zisserman, 2008), Caltech101 (Fei-Fei et al., 2004), OxfordPets (Parkhi et al., 2012), StandfordCars (Krause et al., 2013), UCF101 (Soomro et al., 2012), FGVCAircraft (Maji et al., 2013), Food101

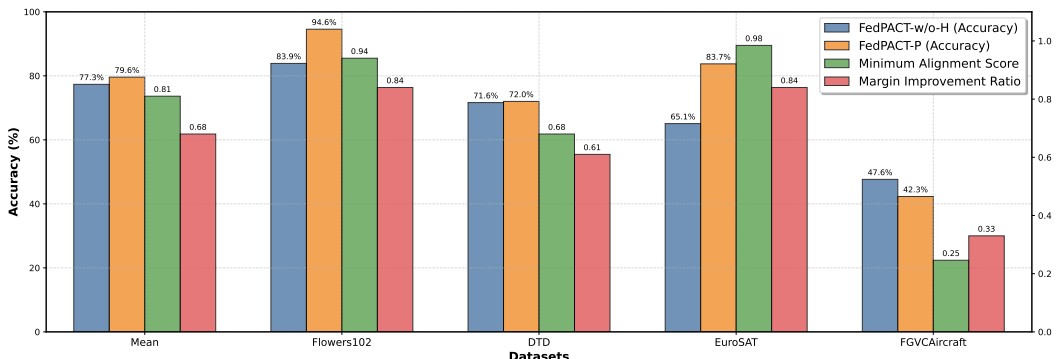

Figure 2: Performance comparison between FedPACT-P and FedPACT-w/o-H with ResNet-50 backbone.

(Bossard et al., 2014), SUN397 (Xiao et al., 2010), DTD (Cimpoi et al., 2014), and EuroSAT (Helber et al., 2019). We adopt a few-shot setting, selecting 1, 4, 8, and 16 training samples per class, and evaluate on the full test sets; client-specific training and test subsets are partitioned via $\text{Dir}(\rho)$.

## 5.2 PERFORMANCE EVALUATION

We evaluate the performance of the proposed method across different datasets and varying levels of data heterogeneity. The experimental results using ResNet-50 and ViT-B/16 backbones are presented in Table 1 and Table 2, respectively. The number of few-shot samples is set to 8. The personalized methods, FedPACT-G, Tip-Adapter, and ECALP, display the average test accuracy across all clients. From the experimental results, it is evident that FedPACT achieves the best performance in both generalization and personalization. Specifically, FedPACT-G shows a significant accuracy improvement over FedPGA and CacheFL, especially under severe data heterogeneity $\rho = 0.1$. FedPACT-G achieves a performance gain of 6 points with ResNet-50 and ViT-B/16. This demonstrates that the contrastive learning approach effectively steers the shared text head towards directions supported by client prototypes, making it more robust to data heterogeneity than aggregation-based knowledge fusion methods. On the other hand, FedPACT-P outperforms the baselines across different backbones and levels of data heterogeneity, indicating that the personalized local prototype cache updates help adapt the model to the local data distribution. The introduction of the shared text head also integrates global knowledge into the local model, enhancing its robustness. Note that, as heterogeneity decreases, FedPACT-P and the two personalized baselines experience a slight drop in overall accuracy. This is because lower heterogeneity implies that clients have more label-rich test sets, which raises the performance requirements during the test phase.

## 5.3 VALIDATION OF THEORETICAL ANALYSIS

In this subsection, we experimentally validate the effectiveness of the proposed server-side contrastive text adaptation and the results of theoretical analysis. First, we define FedPACT without shared text head updates as FedPACT-w/o-H, where each client keeps the initial CLIP text features fixed and trains the local personalized prototype cache. Based on Theorem 3, we define the client prototype alignment score as $\sum_n \frac{\mathbf{v}_{k,n}^\top \bar{\mathbf{v}}_n}{N \|\mathbf{v}_{k,n}\| \|\bar{\mathbf{v}}_n\|}$. In Fig. 2, we present the accuracy of FedPACT-P and FedPACT-w/o-H across different datasets, the minimum client prototype alignment score during the FedPACT-P update process, and the test-set margin improvement ratio for the updated shared text head of FedPACT-P compared to the original text features. The experiments are conducted with an 8-shot setting and $\rho = 0.1$. The experimental results show that on the Flowers102 and EuroSAT datasets, all clients exhibit a high alignment score, greater than 0.9. In these cases, the margin gain of the shared text head can be effectively transferred to the local margin gains of the clients, resulting in a significant accuracy improvement for FedPACT-P compared to FedPACT-w/o-H. Moreover, the margin improvement ratio for the local text sets reaches a high proportion, achieving up to 84%. In contrast, on the FGVCAircraft dataset, some clients exhibit a deviation from the global prototype,

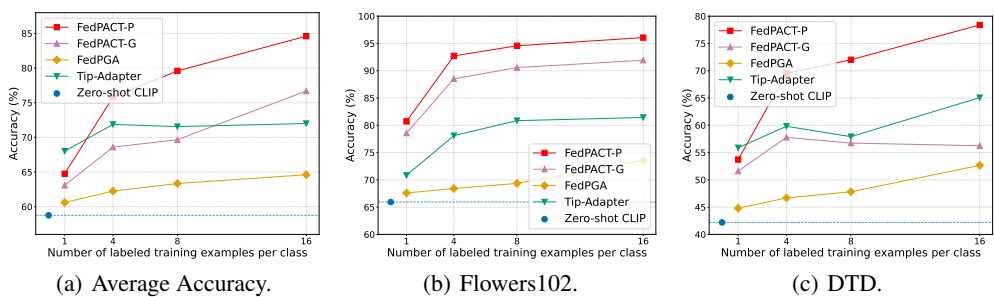

Figure 3: Performance Comparison of FedPACT and baselines under different number of shots per class with ResNet-50 backbone.

with the lowest alignment score being just $0.25$. In this scenario, the margin gain of the shared text head cannot be transferred to the local margin gains, leading to a decrease in accuracy and a lower margin improvement ratio for the local text sets. Thus, while FedPACT-P demonstrates an overall accuracy improvement over FedPACT-w/o-H across multiple datasets, in specific cases such as FGVCAircraft, directly using the original text features yields better model performance, as seen with the FedPACT-w/o-H.

## 5.4 ABLATION STUDY

**Number of Clients $K$:** We test the performance of the proposed FedPACT-G and FedPACT-P on ten datasets with varying numbers of clients, where number of clients are set to $10$, $20$, and $50$, number of few-shot samples is set to $8$, and $\rho = 0.1$, as shown in Table 3. The experimental results indicate that FedPACT consistently delivers significant performance improvements across different number of clients, demonstrating strong robustness to an increasing number of clients. Notably, FedPACT-G maintains an accuracy above $69\%$, indicating its excellent scalability to multiclient federated learning scenarios.

Table 3: Average accuracy (%) of FedPACT-G and FedPACT-P under different number of shots per class with ResNet-50 backbone.

| Num of Clients | $K = 10$ | $K = 20$ | $K = 50$ |
|---|---|---|---|
| CLIP-RN50 | 58.75 | 58.75 | 58.75 |
| FedPGA | 62.37 | 61.75 | 61.49 |
| CacheFL | 61.74 | 61.31 | 60.67 |
| **FedPACT-G** | **69.66** | **69.92** | **69.53** |
| Tip-Adapter | 71.56 | 69.90 | 67.23 |
| ECALP | 56.58 | 52.51 | 41.80 |
| **FedPACT-P** | **79.59** | **79.56** | **74.10** |

**Number of Shots per Class:** We evaluate the performance of the proposed FedPACT-G and FedPACT-P across ten datasets with $[1, 4, 8, 16]$ shots per class and $\rho = 0.1$, as presented in Fig. 3. The experimental results show that FedPACT-P performs lower than Tip-Adapter in the 1-shot setting. However, as the number of shots increases, both FedPACT-G and FedPACT-P achieve significant accuracy improvements over the baselines, with FedPACT-P showing a marked performance gain compared to Tip-Adapter, highlighting the advantages of alternating prototype–text contrastive alignment.

## 6 CONCLUSION

We introduced FedPACT, a parameter-efficient PFL framework based on VLMs that freezes the image encoder, personalizes lightweight personalized prototype caches, and collaborates via a shared text head updated through prototype–text contrastive alignment. We derived a margin-controlled local convergence bound showing that enlarging the feature–text margin tightens convergence bound. We proved that contrastive updates to the shared text head enlarge the global prototype–text margin, and this gain transfers to the local prototype–text margin when the personalized prototypes do not significantly deviate from the global prototypes, thereby improving local convergence. Empirically, FedPACT achieved superior personalized and global accuracy than state-of-the-art methods.

## LLM USAGE STATEMENT

This work was conceived, designed, and executed by the authors. A large language model was used to copy-edit author-written drafts (grammar, phrasing, and minor formatting).

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

# A APPENDIX

## A.1 PROOF OF THEOREM 1

We first refine Assumption 2 using a margin-controlled variance bound, then prove Theorem 1.

**Lemma 4.** *Fix client $k$ and a shared text head $\mathbf{H} = [\mathbf{h}_1; \ldots; \mathbf{h}_N]$. let $g_k$ denote the unbiased mini-batch gradient estimator of $\nabla_{\mathbf{V}}\mathcal{L}_k(\mathbf{V}_k)$ computed from a batch of size $b$. Under the feature $\mathbf{f} = f(x)$ with $\|\mathbf{f}\|_2 = 1$ and the cache logits $S_y = \alpha\varphi(\mathbf{f}\mathbf{v}_{k,j}^\top)\mathbb{1}\{j = y\} + \mathbf{f}\mathbf{h}_y^\top$ with $\varphi(z) = \exp[-\beta(1-z)]$, the mini-batch gradient $g_k$ has bounded variance:*

$$\mathbb{E}\left[\left\|g_k - \nabla_{\mathbf{V}}\mathcal{L}_k(\mathbf{V}_k)\right\|^2\right] \leq \frac{(\alpha\beta)^2(N-1)}{b}\mathbb{E}_{(x,y)\sim\mathcal{D}_k}\left[e^{-m_{\mathbf{H}}(\mathbf{f},y)}\right], \tag{15}$$

*where $m_{\mathbf{H}}(\mathbf{f}, y) := \mathbf{f}^\top\mathbf{h}_y - \max_{j\neq y}\mathbf{f}^\top\mathbf{h}_j$.*

*Proof.* The per-sample cross-entropy gradient w.r.t. the correct-class row $\mathbf{v}_{k,y}$ is

$$\nabla_{\mathbf{v}_{k,y}}\ell_{\mathrm{CE}} = (p_y - 1)\alpha\beta\varphi(\mathbf{f}^\top\mathbf{v}_{k,y})\mathbf{f}, \tag{16}$$

where $p_y$ is the predicted accuracy of class $n$. Then using $\varphi \in (0, 1]$, $p \in [0, 1]$ and $\|\mathbf{f}\| = 1$, we can obtain

$$\|\nabla_{\mathbf{v}_{k,y}}\ell_{\mathrm{CE}}\|^2 \leq (\alpha\beta)^2(1 - p_y). \tag{17}$$

Adding the cache increases the correct-class logit,

$$S_y = \underbrace{\mathbf{f}^\top\mathbf{h}_y}_{\text{text}} + \underbrace{\alpha\varphi(\mathbf{f}^\top\mathbf{v}_{k,y})}_{\text{cache}} > \mathbf{f}^\top\mathbf{h}_y. \tag{18}$$

Since the softmax probability $p_y = \frac{\exp(S_y)}{\sum_j \exp(S_j)}$ is strictly increasing in $S_y$, we have $p_y^{\text{text+cache}} \geq p_y^{\text{text}}$, hence

$$1 - p_y^{\text{text+cache}} \leq 1 - p_y^{\text{text}}. \tag{19}$$

For the text-only logits $z_j = \mathbf{f}^\top\mathbf{h}_j$,

$$1 - p_y^{\text{text}} = \frac{\sum_{j\neq y}e^{z_j - z_y}}{1 + \sum_{j\neq y}e^{z_j - z_y}} \leq (N-1)e^{-m_{\mathbf{H}}(\mathbf{f},y)}. \tag{20}$$

Combining this with $\|\nabla_{\mathbf{v}_{k,y}}\ell_{\mathrm{CE}}\|^2 \leq (\alpha\beta)^2(1 - p_y)$ yields

$$\|\nabla_{\mathbf{v}_{k,y}}\ell_{\mathrm{CE}}\|^2 \leq (\alpha\beta)^2(N-1)e^{-m_{\mathbf{H}}(\mathbf{f},y)}. \tag{21}$$

Define the per-sample gradient

$$g_k(\mathbf{x}) := \nabla_{\mathbf{V}}\mathcal{L}_k(\mathbf{V}_k; f(\mathbf{x}), y), \tag{22}$$

with $(\mathbf{x}, y) \sim \mathcal{D}_k$. The mini-batch estimator is

$$g_k = \frac{1}{b}\sum_{i=1}^{b}g(\mathbf{x}_i), \qquad (\mathbf{x}, y) \overset{\text{i.i.d.}}{\sim} \mathcal{D}_k. \tag{23}$$

Since only the $y$-row of $\nabla_{\mathbf{v}_{k,y}} \ell_{\mathrm{CE}}$ is nonzero, the per-sample gradient matrix has a single nonzero row and hence

$$\|g_k(\mathbf{x})\|^2 = \left\|\nabla_{\mathbf{v}_{k,y}} \ell_{\mathrm{CE}}\right\|^2 \leq (\alpha\beta)^2 (N-1) e^{-m_{\mathbf{H}}(\mathbf{f},y)}. \tag{24}$$

Taking expectation over $(x,y) \sim \mathcal{D}_k$ yields

$$\mathbb{E}\big[\|g(\mathbf{x})\|^2\big] \leq (\alpha\beta)^2 \, (N-1) \, \mathbb{E}_{(x,y)\sim\mathcal{D}_k}\left[e^{-m_{\mathbf{H}}(\mathbf{f},y)}\right]. \tag{25}$$

By independence of the $b$ samples,

$$\mathrm{Var}(g_k) = \frac{1}{b} \mathrm{Var}(g(\mathbf{x})) \ \leq \ \frac{1}{b} \mathbb{E}\big[\|g(\mathbf{x})\|^2\big], \tag{26}$$

where the last step uses $\mathrm{Var}(X) = \mathbb{E}\|X\|^2 - \|\mathbb{E}X\|^2 \leq \mathbb{E}\|X\|^2$. Combining equation 25 and equation 26 gives

$$\mathbb{E}\big[\|g_k - \nabla_{\mathbf{V}} \mathcal{L}_k(\mathbf{V}_k)\|^2\big] = \mathrm{Var}(g_k) \leq \frac{(\alpha\beta)^2(N-1)}{b} \mathbb{E}_{(x,y)\sim\mathcal{D}_k}\left[e^{-m_{\mathbf{H}}(\mathbf{f},y)}\right], \tag{27}$$

which completes the proof. $\qquad\square$

Then, we begin to derive the margin-controlled local convergence. By $L_{\mathbf{V}}$-smoothness of $\mathcal{L}_k$ and the update $\mathbf{V}_k^{(t+1)} = \mathbf{V}_k^{(t)} - \eta g_k^{(t)}$,

$$\mathbb{E}\big[\mathcal{L}_k(\mathbf{V}_k^{(t+1)})\big] \leq \mathbb{E}\big[\mathcal{L}_k(\mathbf{V}_k^{(t)})\big] - \eta \mathbb{E}\big[\|\nabla \mathcal{L}_k(\mathbf{V}_k^{(t)})\|^2\big] + \frac{L_{\mathbf{V}}\eta^2}{2} \mathbb{E}\big[\|g_k^{(t)}\|^2\big]. \tag{28}$$

Decompose $g_k^{(t)} = \nabla \mathcal{L}_k(\mathbf{V}_k^{(t)}) + \xi_k^{(t)}$ with $\xi_k^{(t)} = g_k^{(t)} - \nabla_{\mathbf{V}} \mathcal{L}_k(\mathbf{V}_k^{(t)})$. Then

$$\mathbb{E}\big[\mathcal{L}_k(\mathbf{V}_k^{(t+1)})\big] \leq \mathbb{E}\big[\mathcal{L}_k(\mathbf{V}_k^{(t)})\big] - \left(\eta - \frac{L_{\mathbf{V}}\eta^2}{2}\right) \mathbb{E}\big[\|\nabla \mathcal{L}_k(\mathbf{V}_k^{(t)})\|^2\big] + \frac{L_{\mathbf{V}}\eta^2}{2} \mathbb{E}\big[\|\xi_k^{(t)}\|^2\big]. \tag{29}$$

With $\eta \leq \frac{1}{L_{\mathbf{V}}}$ we have $\eta - \frac{L_{\mathbf{V}}\eta^2}{2} \geq \frac{\eta}{2}$, hence

$$\frac{\eta}{2} \mathbb{E}\big[\|\nabla \mathcal{L}_k(\mathbf{V}_k^{(t)})\|^2\big] \leq \mathbb{E}\big[\mathcal{L}_k(\mathbf{V}_k^{(t)}) - \mathcal{L}_k(\mathbf{V}_k^{(t+1)})\big] + \frac{L_{\mathbf{V}}\eta^2}{2} \mathbb{E}\big[\|\xi_k^{(t)}\|^2\big]. \tag{30}$$

Summing $t = 0$ to $T-1$, we have

$$\min_{0 \leq t < T} \mathbb{E}\big[\|\nabla \mathcal{L}_k(V_k^{(t)})\|^2\big] \leq \frac{2\big(\mathcal{L}_k(\mathbf{V}^{(0)}) - \mathcal{L}_k^{\star}\big)}{\eta T} + L_V \eta \cdot \frac{1}{T} \sum_{t=0}^{T-1} \mathbb{E}\big[\|\xi_k^{(t)}\|^2\big], \tag{31}$$

where $\mathcal{L}_k^{\star} := \inf_{\mathbf{V}} \mathcal{L}_k(\mathbf{V})$ is the optimal objective value for client $k$. By Lemma 4, we have $\mathbb{E}[\|\xi_k\|^2] \leq \frac{(\alpha\beta)^2(N-1)}{b} \mathbb{E}_{(\mathbf{f},y)\sim\mathcal{D}_k}[e^{-m_{\mathbf{H}}(\mathbf{f},y)}]$ and substitute this bound gives

$$\min_{0 \leq t < T} \mathbb{E}\big[\|\nabla_{\mathbf{V}} \mathcal{L}_k(\mathbf{V}_k^{(t)})\|^2\big] \leq \frac{2\big(\mathcal{L}_k(\mathbf{V}^{(0)}) - \mathcal{L}_k^{\star}\big)}{\eta T} + \eta L_{\mathbf{V}} (\alpha\beta)^2 \frac{(N-1)}{b} \mathbb{E}_{(\mathbf{f},y)\sim\mathcal{D}_k}\left[e^{-m_{\mathbf{H}}(\mathbf{f},y)}\right], \tag{32}$$

which completes the proof.

## A.2 PROOF OF LEMMA 2

The gradient of $M(\mathbf{H})$ with respect to $\mathbf{H}$ is:

$$\nabla_{\mathbf{H}} M(\mathbf{H}) = \sum_{k\in\mathcal{K}} \pi_k \sum_{n\in\mathcal{Y}} \nabla_{\mathbf{H}} m_{\mathbf{H}}(\mathbf{v}_{k,n}, n). \tag{33}$$

For each prototype $\mathbf{v}_{k,n}$, the margin gradient $\nabla_{\mathbf{H}} m_{\mathbf{H}}(\mathbf{v}_{k,n}, n)$ is a matrix where only the rows corresponding to the true class $n$ and the hardest negative class $j_{k,n}^* = \arg\max_{j\neq n} \mathbf{h}_j^\top \mathbf{v}_{k,n}$ are non-zero:

$$\frac{\partial m}{\partial \mathbf{h}_j} = \begin{cases} \mathbf{v}_{k,n}, & \text{if } j = n \\ -\mathbf{v}_{k,n}, & \text{if } j = j^* = \arg\max_{j\neq n} \mathbf{h}_j^\top \mathbf{v}_{k,n} \\ \mathbf{0}, & \text{otherwise.} \end{cases} \tag{34}$$

Thus, $\nabla_{\mathbf{H}} M(\mathbf{H})$ is a weighted sum of these sparse gradients. The gradient of $\mathcal{L}_{\text{align}}$ with respect to $\mathbf{h}_j$ is:

$$\frac{\partial \mathcal{L}_{\text{align}}}{\partial \mathbf{h}_j} = \sum_k \frac{\pi_k}{N} \sum_{n \in \mathcal{Y}} \frac{\partial \ell_{k,n}}{\partial \mathbf{h}_j}, \tag{35}$$

where $\ell_{k,n}$ is InfoNCE loss for $\mathbf{v}_{k,n}$. Then, we can obtain

$$\frac{\partial \ell_{k,n}}{\partial \mathbf{h}_j} = \begin{cases} \frac{1}{\tau}(p_{k,n}(j) - 1)\mathbf{v}_{k,n}, & \text{if } j = n \\ \frac{1}{\tau}p_{k,n}(j)\mathbf{v}_{k,n}, & \text{if } j \neq n. \end{cases} \tag{36}$$

Now, $\frac{\partial \mathcal{L}_{\text{align}}}{\partial \mathbf{h}_j}$ can be decomposed into two parts. The first is the positive pull term. This comes from the loss terms where $j = n$, which is

$$-\frac{1}{\tau} \sum_k \frac{\pi_k}{N} \sum_n \mathbb{I}(j = n)(1 - p_{k,n}(j))\mathbf{v}_{k,n}, \tag{37}$$

This term pulls $\mathbf{h}_j$ towards prototypes of class $j$ and aligns with the positive part of $\frac{\partial M}{\partial \mathbf{h}_j}$ (i.e., $\sum_k \pi_k \sum_n \mathbb{I}(j = n)\mathbf{v}_{k,n}$) because both encourage increasing $\mathbf{h}_j^\top \mathbf{v}_{k,j}$ for class $j$. The second is the negative push term. This comes from the loss terms where $j \neq n$, which is

$$\frac{1}{\tau} \sum_k \frac{\pi_k}{N} \sum_n \mathbb{I}(j \neq n)p_{k,n}(j)\mathbf{v}_{k,n}. \tag{38}$$

This term pushes $\mathbf{h}_j$ away from prototypes of other classes and aligns with the negative part of $\frac{\partial M}{\partial \mathbf{h}_j}$ (i.e., $-\sum_k \pi_k \sum_n \mathbb{I}(j = j_{k,n}^*)\mathbf{v}_{k,n}$) when $p_{k,n}(j)$ is large for $j$ being the hardest negative class for some prototype. Specifically, if $j$ is the hardest negative for prototype $\mathbf{v}_{k,n}$, then $p_{k,n}(j)$ is large, and the push term contributes in the direction of $-\mathbf{v}_{k,n}$, which matches the negative part of $\frac{\partial M}{\partial \mathbf{h}_j}$. Thus, overall, $\frac{\partial \mathcal{L}_{\text{align}}}{\partial \mathbf{h}_j}$ is approximately opposite to $\frac{\partial M}{\partial \mathbf{h}_j}$, resulting in

$$\left\langle \frac{\partial M}{\partial \mathbf{h}_j}, \frac{\partial \mathcal{L}_{\text{align}}}{\partial \mathbf{h}_j} \right\rangle < 0, \tag{39}$$

for each $j$. Using the first-order Taylor expansion:

$$M(\mathbf{H}^{(t+1)}) \approx M(\mathbf{H}^{(t)}) + \left\langle \nabla_{\mathbf{H}} M(\mathbf{H}^{(t)}), \mathbf{H}^{(t+1)} - \mathbf{H}^{(t)} \right\rangle_F. \tag{40}$$

Substituting the update $\mathbf{H}^{(t+1)} - \mathbf{H}^{(t)} = -\eta \nabla_{\mathbf{H}} \mathcal{L}_{\text{align}}(\mathbf{H}^{(t)})$, we get:

$$M(\mathbf{H}^{(t+1)}) \approx M(\mathbf{H}^{(t)}) - \eta \left\langle \nabla_{\mathbf{H}} M(\mathbf{H}^{(t)}), \nabla_{\mathbf{H}} \mathcal{L}_{\text{align}}(\mathbf{H}^{(t)}) \right\rangle_F. \tag{41}$$

Since $\left\langle \nabla_{\mathbf{H}} M(\mathbf{H}^{(t)}), \nabla_{\mathbf{H}} \mathcal{L}_{\text{align}}(\mathbf{H}^{(t)}) \right\rangle_F < 0$, it follows that $M(\mathbf{H}^{(t+1)}) > M(\mathbf{H}^{(t)})$ for sufficiently small $\eta$. Thus, the global margin sum $M(\mathbf{H})$ increases monotonically with each gradient descent update.

### A.3 PROOF OF THEOREM 3

Let $\bar{\mathbf{v}}_n = \frac{\sum_{k \in \mathcal{K}} |\mathcal{D}_{k,n}| \hat{\mathbf{v}}_{k,n}}{\sum_{k \in \mathcal{K}} |\mathcal{D}_{k,n}|}$ denote the global prototype for class $n$. The local margin is defined as $m_{\mathbf{H}}(\mathbf{v}_{k,n}, n) = \mathbf{v}_{k,n}^\top \mathbf{h}_n - \max_{j \neq n} \mathbf{v}_{k,n}^\top \mathbf{h}_j$, and let $j^* = \arg\max_{j \neq n} \mathbf{v}_{k,n}^\top \mathbf{h}_j$ be the hardest negative class. As shown in equation 34, the gradient of $m$ with respect to $\mathbf{H}$ is given by:

$$\nabla_{\mathbf{H}} m_{\mathbf{H}}(\mathbf{v}_{k,n}, n) = [\mathbf{0}, \dots, \mathbf{v}_{k,n}, \dots, -\mathbf{v}_{k,n}, \dots, \mathbf{0}]^\top, \tag{42}$$

where the only non-zero rows correspond to the true class $n$ and the hardest negative class $j^*$.

Applying a first-order Taylor expansion, the margin at the next iteration can be approximated as:

$$m_{\mathbf{H}^{(t+1)}}(\mathbf{v}_{k,n}, n) \approx m_{\mathbf{H}^{(t)}}(\mathbf{v}_{k,n}, n) - \eta \left\langle \nabla_{\mathbf{H}} m_{\mathbf{H}^{(t)}}(\mathbf{v}_{k,n}, n), \nabla_{\mathbf{H}} \mathcal{L}_{\text{align}}(\mathbf{H}^{(t)}) \right\rangle_F. \tag{43}$$

Hence, for a sufficiently small learning rate $\eta$, the margin increases if the Frobenius inner product is negative. This inner product expands to:

$$\langle \nabla_{\mathbf{H}} m, \nabla_{\mathbf{H}} \mathcal{L}_{\text{align}} \rangle_F = \mathbf{v}_{k,n}^{\top} \frac{\partial \mathcal{L}_{\text{align}}}{\partial \mathbf{h}_n} - \mathbf{v}_{k,n}^{\top} \frac{\partial \mathcal{L}_{\text{align}}}{\partial \mathbf{h}_{j^*}}. \tag{44}$$

Under the condition that $\frac{\mathbf{v}_{k,n}^{\top} \bar{\mathbf{v}}_n}{\|\mathbf{v}_{k,n}\| \|\bar{\mathbf{v}}_n\|} > \varepsilon$, the prototype $\mathbf{v}_{k,n}$ is highly aligned with the global prototype $\bar{\mathbf{v}}_n$ and, by the separation of different class prototypes, is distant from prototypes of other classes. We now analyze the two terms in the inner product.

First, consider the gradient of the loss with respect to the true class embedding:

$$\frac{\partial \mathcal{L}_{\text{align}}}{\partial \mathbf{h}_n} = -\frac{1}{\tau N} \sum_{k'} \pi_{k'} (1 - p_{k',n}(n)) \mathbf{v}_{k',n} + \frac{1}{\tau N} \sum_{k'} \pi_{k'} \sum_{n' \neq n} p_{k',n'}(n) \mathbf{v}_{k',n'}. \tag{45}$$

The product $\mathbf{v}_{k,n}^{\top} \frac{\partial \mathcal{L}_{\text{align}}}{\partial \mathbf{h}_n}$ is dominated by the first term, which is large and negative due to the high similarity $\mathbf{v}_{k,n}^{\top} \mathbf{v}_{k',n}$ and the fact that $(1 - p_{k',n}(n)) > 0$. The second term is small and positive because both the probability $p_{k',n'}(n)$ and the inter-class similarity $\mathbf{v}_{k,n}^{\top} \mathbf{v}_{k',n'}$ are low for $n' \neq n$.

Next, consider the gradient with respect to the hardest negative class embedding:

$$\frac{\partial \mathcal{L}_{\text{align}}}{\partial \mathbf{h}_{j^*}} = -\frac{1}{\tau N} \sum_{k'} \pi_{k'} \sum_{n'} \mathbb{I}(j^* = n')(1 - p_{k',n'}(j^*)) \mathbf{v}_{k',n'}$$

$$+ \frac{1}{\tau N} \sum_{k'} \pi_{k'} \sum_{n'} \mathbb{I}(j^* \neq n') p_{k',n'}(j^*) \mathbf{v}_{k',n'}. \tag{46}$$

The product $\mathbf{v}_{k,n}^{\top} \frac{\partial \mathcal{L}_{\text{align}}}{\partial \mathbf{h}_{j^*}}$ is dominated by the part of the second summation where $n' = n$. This contribution is large and positive because: (i) $p_{k',n}(j^*)$ is high since $j^*$ is the hardest negative class for prototypes of class $n$, and (ii) the similarity $\mathbf{v}_{k,n}^{\top} \mathbf{v}_{k',n}$ is high. The first term in the gradient is small and negative since the similarity $\mathbf{v}_{k,n}^{\top} \mathbf{v}_{k',j^*}$ is low for $j^* \neq n$.

Therefore, $\mathbf{v}_{k,n}^{\top} \frac{\partial \mathcal{L}_{\text{align}}}{\partial \mathbf{h}_n}$ is a negative value, while $\mathbf{v}_{k,n}^{\top} \frac{\partial \mathcal{L}_{\text{align}}}{\partial \mathbf{h}_{j^*}}$ is a positive value. Consequently, their difference is negative for a sufficiently high $\varepsilon$, ensuring the inner product is negative. It follows that $m_{\mathbf{H}^{(t+1)}}(\mathbf{v}_{k,n}, n) > m_{\mathbf{H}^{(t)}}(\mathbf{v}_{k,n}, n)$ for a sufficiently small $\eta$.

### A.4 FULL RESULTS OF ABLATION STUDY ON THE NUMBER OF SHOTS PER CLASS

As shown in Fig. 4, we present the full results of the ablation study on the number of shots per class. It can be observed that, except for the Food101 datasets where the performance of FedPACT-G is lower than the baseline, the accuracy of the other datasets significantly exceeds the baseline as the number of shots increases. Notably, on the FGVCAircraft dataset, while FedPACT performs worse than both baselines at lower numbers of shots, its accuracy significantly improves and surpasses the baselines when the number of shots reaches 16. This indicates that the prototype-text contrastive alignment method can effectively learn knowledge from datasets with larger amounts of data, thereby enhancing model performance.

### A.5 EXPERIMENTAL RESULTS USING GPT3-GENERATED PROMPTS

We initialize the shard text head with superior text features (Zhang et al., 2024), where the class-specific prompts are generated by GPT-3. The experimental results using ResNet-50 and ViT-B/16 backbones are presented in Table 4 and Table 5, respectively. The number of few-shot samples is set to 8. Experimental results show that, although the accuracy of FedPACT-G and FedPACT-P still surpasses the baselines on most datasets, there is no significant performance improvement compared to Table 1 and Table 2 when using this improved initialization. This indicates that the prototype-text contrastive alignment method itself is robust and effective in optimizing the model, and therefore, its performance does not see significant improvement with better initialization.

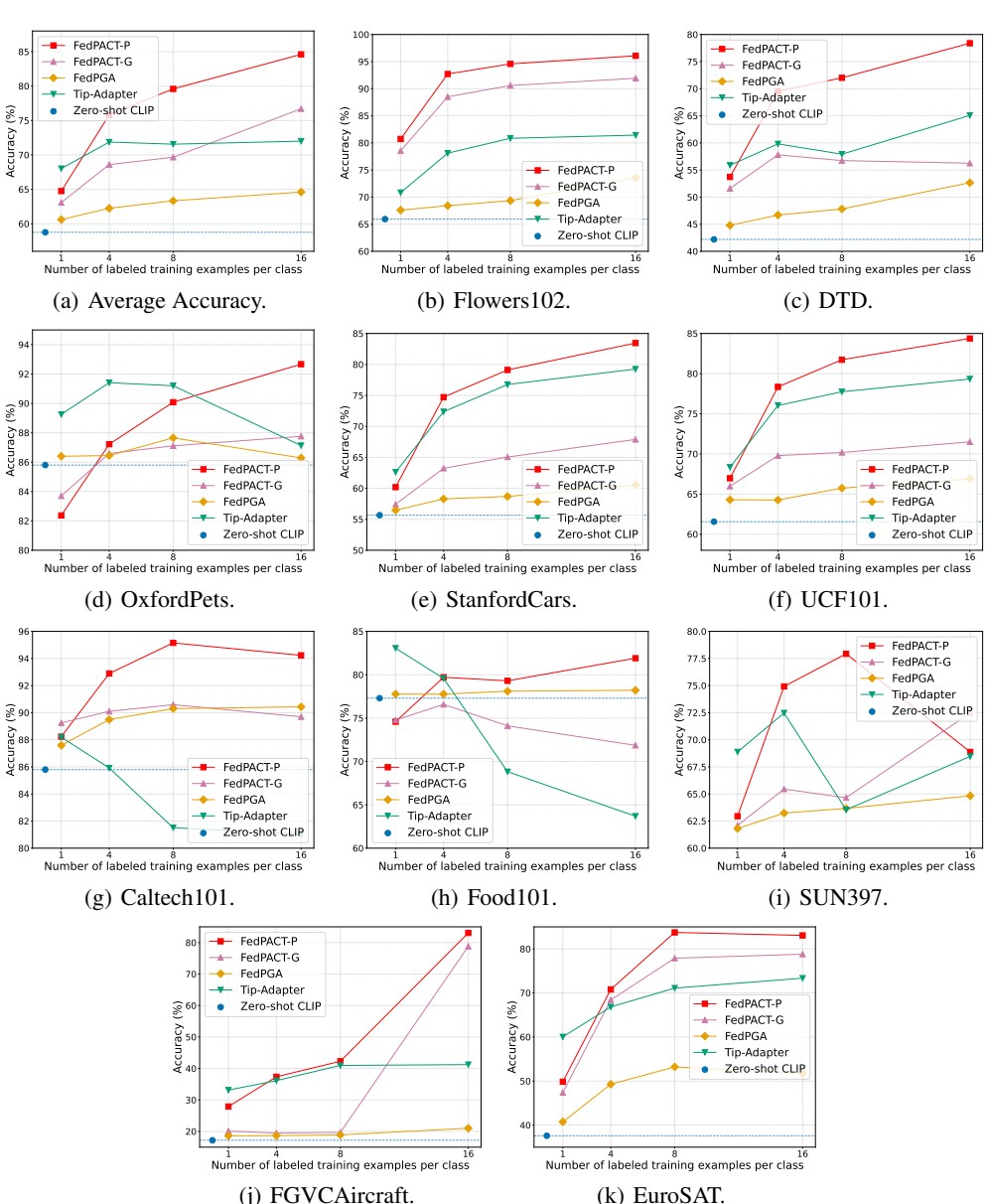

Figure 4: Performance Comparison of FedPACT and baselines under different number of shots per class with ResNet-50 backbone.

Table 4: Test accuracy (%) for FedPACT and baselines with ResNet-50 backbone and GPT3-generated prompts.

| Non-IID ($\rho$) | Method | Flower | DTD | Pets | Cars | UCF | Caltech | Food | SUN | Aircraft | EuroSAT | Mean |
|---|---|---|---|---|---|---|---|---|---|---|---|---|
| | CLIP-RN50 | 66.22 | 48.64 | 85.04 | 57.37 | 58.58 | 89.57 | 76.99 | 62.63 | 19.44 | 40.10 | 60.45 |
| $\rho = 0.1$ | FedPGA | 70.48 | 54.08 | 86.48 | 60.53 | 63.39 | 90.91 | **77.70** | 65.89 | **21.75** | 50.12 | 64.13 |
| | CacheFL | 69.27 | 51.60 | 86.26 | 59.28 | 63.79 | **91.32** | 77.57 | 64.74 | 20.70 | 44.96 | 62.94 |
| | **FedPACT-G** | **90.17** | **59.28** | **87.11** | **65.13** | **70.42** | 91.20 | 74.27 | 64.84 | 19.65 | **79.27** | **70.13** |
| | Tip-Adapter | 81.14 | 60.90 | **90.24** | 77.49 | 78.21 | 83.65 | 68.42 | 70.39 | **42.42** | 75.28 | 72.81 |
| | ECALP | 61.59 | 51.44 | 64.92 | 56.24 | 59.29 | 61.65 | 58.33 | 59.10 | 39.52 | 50.55 | 56.26 |
| | **FedPACT-P** | **94.44** | **73.25** | 89.58 | **79.36** | **81.00** | **95.62** | **78.53** | **78.28** | 41.89 | **82.66** | **79.46** |
| $\rho = 0.3$ | FedPGA | 73.45 | 56.97 | **87.14** | 63.08 | 66.06 | 90.91 | **78.14** | 66.86 | 22.50 | 53.58 | 65.86 |
| | CacheFL | 72.27 | 52.90 | 86.59 | 60.48 | 67.17 | **91.32** | 77.89 | 66.41 | 21.90 | 51.86 | 64.87 |
| | **FedPACT-G** | **89.77** | **59.40** | 86.94 | **65.66** | **70.76** | 91.28 | 74.83 | 65.72 | **23.01** | **78.91** | **70.62** |
| | Tip-Adapter | 72.68 | 51.79 | 88.64 | 68.75 | 69.82 | 82.52 | 63.03 | 61.97 | 31.16 | 64.02 | 65.43 |
| | ECALP | 40.06 | 36.55 | 44.17 | 40.79 | 39.79 | 50.44 | 45.39 | 41.84 | 27.19 | 42.84 | 40.90 |
| | **FedPACT-P** | **92.70** | **66.36** | **88.88** | **72.70** | **76.32** | **92.78** | **77.41** | **72.84** | **33.41** | **80.94** | **75.43** |

Table 5: Test accuracy (%) for FedPACT and baselines with ViT-B/16 backbone and GPT3-generated prompts.

| Non-IID ($\rho$) | Method | Flower | DTD | Pets | Cars | UCF | Caltech | Food | SUN | Aircraft | EuroSAT | Mean |
|---|---|---|---|---|---|---|---|---|---|---|---|---|
| | CLIP-ViTB/16 | 72.88 | 53.31 | 90.30 | 66.04 | 67.04 | 94.24 | 85.87 | 67.95 | 27.33 | 54.25 | 67.92 |
| $\rho = 0.1$ | FedPGA | 77.26 | 56.38 | 91.06 | 69.36 | 71.98 | 95.01 | **86.34** | 71.38 | **31.26** | 57.23 | 70.72 |
| | CacheFL | 75.23 | 55.56 | 90.84 | 67.17 | 72.77 | **95.17** | 86.23 | 70.50 | 28.41 | 55.79 | 69.76 |
| | **FedPACT-G** | **94.28** | **64.78** | **92.42** | **74.31** | **77.93** | 94.24 | 85.38 | 70.70 | 26.46 | **79.72** | **76.02** |
| | TAdapter | 85.46 | 59.77 | 93.19 | 84.11 | 84.72 | 83.32 | 74.36 | 72.06 | 51.62 | 79.58 | 76.81 |
| | ECALP | 63.23 | 51.82 | 64.72 | 62.77 | 64.40 | 62.69 | 64.03 | 62.74 | 47.86 | 54.20 | 59.84 |
| | **FedPACT-P** | **96.13** | **75.81** | **94.01** | **85.35** | **86.43** | **97.49** | **87.81** | **82.01** | **52.08** | **83.50** | **84.06** |
| $\rho = 0.3$ | FedPGA | 81.00 | 60.28 | 91.39 | 71.60 | 74.15 | 95.90 | 86.49 | **72.18** | **31.41** | 58.79 | 72.31 |
| | CacheFL | 79.58 | 56.26 | 91.66 | 69.11 | 75.50 | **95.94** | 86.48 | 71.80 | 30.15 | 58.51 | 71.49 |
| | **FedPACT-G** | **94.28** | **65.84** | **92.29** | **73.66** | **77.50** | 94.44 | 85.79 | 71.73 | 30.96 | **79.84** | **76.63** |
| | TAdapter | 78.40 | 50.54 | 92.49 | 76.67 | 77.99 | 83.27 | 71.23 | 63.55 | 38.74 | 68.45 | 70.13 |
| | ECALP | 40.47 | 37.92 | 46.33 | 46.64 | 42.77 | 51.20 | 49.46 | 45.05 | 35.37 | 42.45 | 43.76 |
| | **FedPACT-P** | **95.23** | **71.18** | **93.88** | **80.14** | **82.10** | **96.24** | **86.95** | **77.61** | **44.56** | **81.84** | **80.97** |

