# OpenReview forum: "Personalized Federated Adaptation via Prototype–Text Contrastive Alignment"
_ICLR.cc/2026/Conference — Submitted to ICLR 2026_

### Official Review · Reviewer_McFP · 2025-10-30

**Soundness:** 3
**Presentation:** 3
**Contribution:** 2
**Rating:** 2
**Confidence:** 4

**Summary:**

The paper introduces a novel personalized federated learning method, FedPACT (Federated Adaptation via Prototype–Text Contrastive Alignment), which addresses key challenges in federated learning: data heterogeneity and communication overhead. It proposes a framework where clients personalize a lightweight prototype cache and the server maintains a shared text head for global information aggregation. FedPACT minimizes communication costs and enhances personalization by aligning prototype-based and text-embedded representations. The authors theoretically analyze the convergence of their method and demonstrate its effectiveness through experimental results, showing improved performance in personalization and global generalization over existing methods.

**Strengths:**

1. The paper successfully tackles two critical challenges in federated learning: data heterogeneity and communication overhead. The FedPACT framework reduces communication costs and mitigates the negative impact of data heterogeneity by using a shared text embedding and client-specific prototypes, trained separately on the server and client sides.

2. The authors provide a detailed theoretical analysis of FedPACT's convergence properties. They demonstrate that the proposed contrastive alignment method leads to improved convergence, highlighting the advantage of enlarging the prototype–text margin to enhance client model alignment with the global model.

3. The writing is clear, well-structured, and easy to follow, making complex concepts accessible. The methodology is explained with sufficient detail, and the results are presented in a manner that makes the findings comprehensible to the reader.

**Weaknesses:**

1. The paper tests the FedPACT-G (global model) and FedPACT-P (personalized model) separately, but it doesn't investigate how the combination of these two components could improve performance. The separation of these two components makes it unclear how they relate to each other and whether integrating them would yield better results.

2. The comparison with other federated learning methods, particularly those related to federated prompt learning (e.g., Harmonizing Generalization and Personalization in Federated Prompt Learning, Mixture of Experts Made Personalized: Federated Prompt Learning for Vision-Language Models), is minimal. Including these baseline methods would have provided a clearer picture of how FedPACT compares with emerging methods in the field.

3. The paper focuses on reducing communication overhead but lacks an in-depth experimental analysis comparing the communication costs of FedPACT with other methods. A comparison of communication overhead across methods would strengthen the claim of FedPACT’s efficiency.

4. While the paper addresses label shift, it does not explore domain shift, which is another type of data heterogeneity. Including experiments on domain shift would provide a more comprehensive evaluation of FedPACT's robustness.

**Questions:**

1. Would combining FedPACT-G and FedPACT-P by weight-averaging or integrating their outputs improve the overall performance? What would the effect be on model performance?

2. The current method updates the text embedding only on the server-side. How would the performance change if training text prompts at the client-side instead as a personalized part? Could this improve personalization, especially for clients with highly specific data?

---

### Official Review · Reviewer_5tPi · 2025-10-30

**Soundness:** 3
**Presentation:** 3
**Contribution:** 3
**Rating:** 6
**Confidence:** 3

**Summary:**

FedPACT summarized that the two key challenges in current FL are data heterogeneity and high communication and computational overhead. Therefore, they came up with the idea of combining VLMs to anchor semantic spaces using pre-trained language as a stable reference across clients, thereby achieving efficient individualization without updating the backbone. This article has conducted sufficient theoretical analysis and experiments on multiple datasets to demonstrate the effectiveness of the method.

**Strengths:**

1.Very thorough theoretical analysis and experimental verification, with very convincing results that make me believe in the effectiveness of the method.
2.The design of this article only requires the transmission of prototype cache and text head, which indeed alleviates the problem of communication overhead.

**Weaknesses:**

1.The assumption in theoretical analysis is that the client prototype is highly aligned with the global prototype, but when extremely Non-IID or the number of clients is extremely large, this condition may not hold, resulting in incomplete realization of theoretical gains, and convergence assumptions may be difficult to fully satisfy in practice.

2.In fact, the research on related work in this article is not sufficient. There are already highly similar methods for insight, such as the one I found [1] which shares the same viewpoint. I hope the author can fully investigate related work with similar insights and make sufficient comparisons with SOTA in experiments.

3.Although emphasizing the low communication and training costs, and I may also be willing to accept this conclusion, stronger evidence should appear in the experiments. I hope the author can provide sufficient discussion on communication costs, storage costs, and computational costs.

[1]Wu X, Niu J, Liu X, et al. Enhancing Visual Representation with Textual Semantics: Textual Semantics-Powered Prototypes for Heterogeneous Federated Learning[J]. arXiv preprint arXiv:2503.13543, 2025.

**Questions:**

1.The paper mainly focuses on the heterogeneity of class, but does not discuss domain shift. If the client data comes from different visual domains (such as natural images, sketches, medical images), can FedPACT still effectively align? How to address subtle semantic differences between different domain sources?
2.Although there is a significant reduction in communication overhead during training, in actual deployment, the inference phase requires calling both the prototype cache and the text header (if I understand correctly). Will this result in additional latency or memory usage? Is there a delay test?

---

### Official Review · Reviewer_iyYE · 2025-10-31

**Soundness:** 3
**Presentation:** 3
**Contribution:** 3
**Rating:** 4
**Confidence:** 5

**Summary:**

The paper proposes FedPACT, a parameter-efficient PFL framework that freezes the pretrained image encoder of a VLM (e.g., CLIP) and trains only two light modules: a server-side shared text head and client-side personalized prototype caches. Training alternates between (i) clients optimizing local prototypes with the shared text head fixed and (ii) the server updating the text head via prototype–text contrastive learning (prototype with its class text as positive; other classes as negatives). Inference sums the residual scores from text head and prototypes. The authors provide a margin-controlled local convergence bound and prove that contrastive updates strictly increase global prototype–text margin and can transfer to local margins when prototypes align with global centroids. Across 10 datasets, ResNet-50/ViT-B/16 backbones, and few-shot non-IID settings (ρ=0.1, 0.3), FedPACT outperforms strong baselines in both global generalization and personalization.

**Strengths:**

1. New paradigm: shared text head as a semantic anchor reduces sensitivity to non-IID aggregation.
2. Efficiency: freezing the visual backbone and updating small heads/caches suggests lower communication and compute than full-model aggregation.
3. Theory–practice bridge: margin-controlled convergence and global-to-local transfer provide mechanism-level clarity.
4. Comprehensive evaluation: multiple backbones/datasets/ρ/K/shots, including analysis of 1-shot vs. higher-shot regimes.

**Weaknesses:**

1. Lack of comm/compute metrics: despite “parameter-efficient” claims, the paper does not report bytes/round, latency, or FLOPs vs. baselines; such plots would strengthen the efficiency claim.
2. Prompt sensitivity: GPT-3-generated prompts are used—please include robustness analyses w.r.t. templates/languages.
3. Broader generalization: experiments focus on image classification; cross-domain or cross-modal validations would improve external validity.
4. In terms of theoretical proof of margin-controlled local convergence bound, there is no experimental verification.

**Questions:**

1. Communication footprint: What are per-round upload/download sizes of the text head and prototypes? How do they compare to FedPGA/CacheFL?
2. Prompt robustness: How sensitive are results to GPT-3 prompts or manually designed prompts? Please provide sensitivity curves/ablations.
3. Missing-class clients: Under extreme class sparsity or unseen classes, does the prototype–text contrast remain stable? Any strategy for dynamic class expansion or temperature adaptation?
4. Alignment threshold ε: Practically, how do you estimate/monitor prototype–centroid alignment to trigger early stopping or resampling?

---

### Official Review · Reviewer_BKZz · 2025-11-02

**Soundness:** 2
**Presentation:** 1
**Contribution:** 2
**Rating:** 0
**Confidence:** 3

**Summary:**

The paper proposes an algorithm (FedPACT) for personalized Federated Learning of Vision-Language Models. FedPACT maintains a shared text head at the server, and client-specific prototype caches as the only trainable parameters while the image encoder of the VLM stays frozen. The shared text head is updated with contrastive learning using text of correct vs incorrect class labels. A theoretical analysis is presented for convergence rate of the norm of gradient of personalized prototype cache loss w.r.t. the negative exponent of the prototype-text margin (given a particular shared text head). Experimental results are provided on both personalization and generalization accuracy of FedPACT and give good results under data heterogeneity.

**Strengths:**

(S1) The paper works on how to adapt/personalize the text encoder of VLMs in the Federated learning setting where clients have non-IID data. This is a significant question since there is an increasing willingness in industry to deploy VLMs closer to the edge (including on device) where data heterogeneity challenges exist.

(S2) The theorem statements look sensible and intuitive. The convergence rate is being analyzed by upper bounding in terms of a decreasing function of the prototype-text margin. The contrastive loss is designed to improve this margin.

**Weaknesses:**

The presentation quality is extremely poor. Since evaluating all other aspects of the work relies on the reader/reviewer being able to understand the problem being addressed, I am certain that there are good ideas in the paper that I was simply unable to spot/understand.

(W1) The introduction, as it is currently written, reads like a solution that is in search of a problem. It is unclear what problem is being motivated that will subsequently be solved and what type of empirical evidence will be provided. Note that VLMs are used for a wide variety of tasks and are applicable is different scenarios in different ways. Not every scenario requires federated learning or personalization.

(W2) The introduction uses critical jargon that is not properly defined, and important specifics of the federated learning setting are missing making the paper hard to follow. What is a prototype? Is the cross-device or cross-silo setting being addressed? What constraints on privacy are being considered for the algorithmic developments in the paper?

(W3) Contextualization of the contributions of the paper w.r.t. related work section is unclear. The related work section is currently written as a laundry list without adequate commentary or highlighting of specific gaps that will be addressed by the techniques being introduced in the paper. Further, the paper seems to have missed a couple of highly relevant recent works - [A] Zhang, J., Liu, Y., Hua, Y., & Cao, J. (2024). FedTGP: Trainable Global Prototypes with Adaptive-Margin-Enhanced Contrastive Learning for Data and Model Heterogeneity in Federated Learning. Proceedings of the AAAI Conference on Artificial Intelligence, 38(15), 16768-16776. (https://arxiv.org/abs/2401.03230), and [B] Zhou, Y., Qu, X., You, C., Zhou, J., Tang, J., Zheng, X., Cai, C., & Wu, Y. (2025). FedSA: A Unified Representation Learning via Semantic Anchors for Prototype-based Federated Learning. Proceedings of the AAAI Conference on Artificial Intelligence, 39(21), 23009-23017. (https://doi.org/10.1609/aaai.v39i21.34464)

(W4) The contributions don't talk about experimental results at all even though results are being presented in the paper. Lines 62-63 talk about reducing communication and computation costs but I could not find where corresponding evidence is provided in the paper that compares on computation and communication cost metrics.

(W5) Lines 68-69, 409-410 talk about prototype-text contrastive alignment as being introduced to mitigate data heterogeneity. There is inadequate commentary/intuition provided on why FedPACT is able to deal with data heterogeneity better than the other aggregation strategies/algorithms that the paper compares against. This requires more granular analysis to conclude. Results in Table 1 clearly indicate that FedPACT is not always the best at generalization accuracy conditioned on the dataset. Also, confidence intervals are missing.

(W6) There seems to be a confusion on what is an ablation study. The results in Section 5.4 are testing robustness to the pFL system's hyperparameters. These are not ablation studies. Lines 420-422 describe FedPACT-w/o-H which would be a candidate for ablation study.

**Questions:**

Could the authors address the extensive list of weaknesses? Most important of these are (W1) through (W5).

---

### Meta-Review · Area_Chair_M97r · 2026-01-06

**Summary:**

The reviewer's concerns are mostly on presentation quality, robustness and broader generalization, and experimental analysis.

**Reviewer Concerns:**

No rebuttal is provided therefore no score needs to be changed.

**Reviewer Scores:**

None.

---

### Decision · Program_Chairs · 2026-01-26

Reject